# Protein-Ligand Docking Surrogate Models: A SARS-CoV-2 Benchmark for Deep Learning Accelerated Virtual Screening

**Austin Clyde, Thomas Brettin, Alexander Partin, Hyunseung Yoo,**
**Yadu Babuji, Ben Blaiszik, Arvind Ramanathan, Rick Stevens**
Data Science and Learning Division
Argonne National Laboratory
Lemont, IL 60439, USA

**Andre Merzky, Matteo Turilli, Shantenu Jha**
Computational Sciences Initiative
Brookhaven National Laboratory
Upton, New York 11973, USA

## Abstract

We propose a benchmark to study surrogate model accuracy for protein-ligand docking. We share a dataset consisting of 200 million 3D complex structures and 2D structure scores across a consistent set of 13 million "in-stock" molecules over 15 receptors, or binding sites, across the SARS-CoV-2 proteome. Our work shows surrogate docking models have six orders of magnitude more throughput than standard docking protocols on the same supercomputer node types. We demonstrate the power of high-speed surrogate models by running each target against 1 billion molecules in under a day (50k predictions per GPU seconds). We showcase a workflow for docking utilizing surrogate ML models as a pre-filter. Our workflow is ten times faster at screening a library of compounds than the standard technique, with an error rate less than 0.01% of detecting the underlying best scoring 0.1% of compounds. Our analysis of the speedup explains that to screen more molecules under a docking paradigm, another order of magnitude speedup must come from model accuracy rather than computing speed (which, if increased, will not anymore alter our throughput to screen molecules). We believe this is strong evidence for the community to begin focusing on improving the accuracy of surrogate models to improve the ability to screen massive compound libraries 100x or even 1000x faster than current techniques.

## 1 Introduction

Viral pandemics, antibiotic resistant bacteria, or fungal infections such as *candida auris* are fundamental threats to human health [3, 13]. SARS-CoV-2 (COVID-19) shocked the world with its first appearance estimated in the Fall/Winter of 2019 to becoming a global crisis by March, 2020 when it was declared a global pandemic by the World Health Organization. We believe the ML community can help in an effort for global preparedness by developing computational infrastructure to scale computational drug discovery efforts. Extremely fast and high throughput computational techniques can be used at the beginning of pandemics or even as surveillance systems are screening billions of compounds against entire protemes to find the most promising leads.

Preprint. Under review.

In response to the COVID-19 pandemic, scientists across the globe began a massive drug discovery effort spanning traditional targeted combinatorial library screening [22, 15, 6], drug repurposing screens [1], and crowd-sourced community screening [2]. In common to all these efforts was the ability to leverage off-the-shelf molecular docking programs rapidly. Molecular docking programs are essential preliminary drug discovery efforts as they predict the structure drug candidates form in complex with the protein targets of focus.

The first phase of computational drug discovery often starts with identifying regions of a protein (receptors) that are reasonable targets for small molecule binding, followed by searching small molecules for their ability to bind the receptor. These receptor-binding computations are performed using standard docking software, such as DOCK , AutoDock , UCSF Dock , and many others [28]. These molecular docking programs perform a search in which the conformation of the ligand in relation to the receptor (pose) is evaluated recursively until the convergence to the minimum energy is reached. An affinity score, $\Delta G$ (U total in kcal/mol), is then used to rank candidate poses. Protein-ligand docking software has two main outputs: a pose (ligand conformer in a particular complex with the protein and an associated score. Scores are approximates to the binding free energy, though the units and interpretation of scores depend on the exact docking protocol used. The inputs are an ensemble of 3D ligand conformations and a prepared protein receptor. We call a protein receptor a protein structure that has been annotated with a particular binding pocket coordinates (i.e. the binding site box).

Several open databases of commercially available compounds, not yet commercially available but with synthesizable properties, and compounds that have an unknown synthesizability are available to the scientific community. Some of the largest include ZINC , Enamine Real , GDB-13 and SAVI with each containing on order $10^9$ compounds. Recently, *Babuji* released an aggregate collection of over $10^9$ compounds in representations suitable for deep learning [4].

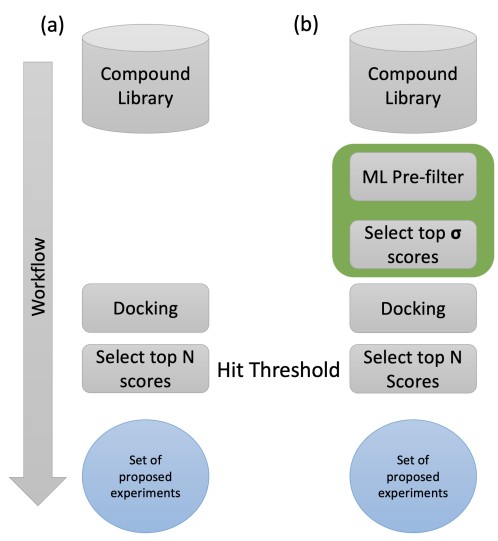

Figure 1: **Overview of Surrogate Prefilter Then Dock.** (a) This workflow is the standard virtual screening workflow consisting of taking a compound library, docking the library, and selecting the top scoring compounds. (b) This workflow is the surrogate prefilter and dock (SPFD) workflow where there are two thresholds, both the hit threshold as in (a) but also the $\sigma$ threshold which decides how many compounds to pass through to docking. Ultimately, in order to not miss the few leads that would normally pass to experiments, our evaluation technique aims to ensure that the set of experiments out of both workflows matches.

These representations included drug descriptors, fingerprints, images and canonical smiles. Searching collections of this size using traditional docking tools is not practical.

In this work, we demonstrate deep learning to accelerate docking, thereby expanding our ability search space of small molecules. We do so by first illustrating our large-scale effort of training surrogate models across the SARS-CoV-2 proteome. We illustrate the speed of ML-based surrogate docking models by performing inference on over 1 billion compounds in hours on supercomputer GPU nodes. ML-based surrogate docking models can be combined with classical docking through a prefilter regime where molecules are first screened with the fast surrogate model and the top scoring products are passed to traditional docking (Fig. 1). We call this technique surrogate model prefilter then dock (SPFD). We perform a detailed analysis of SPFD by combining model accuracy performance, computational throughput, and virtual screening detection sensitive into a single analysis framework. Our analysis framework indicates our workflow is 10x as fast as the traditional only docking method with less than 0.1% error of detecting top hits. The framework indicates our ability to improve this speedup is mainly limited by surrogate model accuracy rather than computing power. Our hope is

this benchmark allows the community to make progress on the modeling aspect, leading to tangible 100x or even 1000x improvement in throughput for virtual ligand screening campaigns.

## 2   Related Work

Virtual screening is a broad category of computational techniques for searching databases of compounds to locate an exciting subset of leads for downstream tasks [23]. The goal of virtual screening is to propose molecules for testing in biological assays. Generally, the space of compounds available is many orders of magnitude beyond what is feasible for wet-lab testing. Standard practice in virtual screening is to utilize molecular docking. Molecular docking is a computational technique for simulating various poses a ligand can take in a protein binding pocket, returning the likely poses, and scoring those poses based on the favorability of the energetics [8].

From an algorithmic perspective, molecular docking is a computational means of assigning a favorability score to a molecule for a particular protein pocket. Various groups work with molecular docking at scale to discover new chemotypes or potent therapeutics [17, 12, 21]. Recently, more groups have begun to develop machine learning models re-score the scores that come from docking (i.e., connecting the computational scores from a model of energetics to success in in-vitro assays) [20, 26]. Other groups have tried to address the computational throughput problem of docking [27, 10]. Researchers have attempted to redesign docking protocols to run on GPUs or to reduce the complexity of docking through further approximations [16, 11].

In general, we have found the adoption of ML techniques to be a difficult task in the biological sciences [30]. To the best of our knowledge, no benchmark or public dataset release focuses on bridging the gap between the virtual screening community and the machine learning community. In this paper, the benchmark we outline attempts to bridge this gap. We propose utilizing surrogate models as a pre-filter before docking. We believe this will increase the potential adoption of ML in biological research as (1) it reduces any epistemic reliance on the surrogate models and (2) directly addresses the communities' current problem of interest, which is expanding chemical library size to go beyond the standard molecule libraries which have been screened time and time again for the past 30 years [24]. Our proposal for surrogate models as a pre-filter solves an epistemic problem since the ML model is designed to filter a sizeable molecular library down to a computationally tractable library size. The virtual screening community can continue their standard practice of docking on (or whichever downstream protocol they desire). This positions any gains from the ML research community as a starting point for the drug discovery community rather than a middle-man where epistemic reliance may be required. Second, our benchmark proposal focuses directly on expanding the tractability of computing large library sizes, which has been an impetus in drug discovery. Together, we believe this situates our benchmark apart from currently published benchmarks and works towards a fundamental problem of building bridges between communities. Furthermore, this motivates a novel mode of statistical thinking for ML researchers. Instead of focusing on the central tendency of the dataset, this regression context of predicting computational scores requires studying how to detect very small subsets of a library (0.01% or less) with near perfect precision.

## 3   Dataset Overview

As part of our drug discovery campaign for SARS-CoV-2 [6], we developed a database of docked protein-ligand across 15 protein targets and 12M compounds as well as the complexes' associated scores (see SI.7.3 for docking pipeline details).

The database contains two related tasks. The first is predicting a ligand's docking score to a receptor based on 2D structural information from the ligand, and the second is a pan-receptor model which encodes the protein target in order to use a single model across different ligands and different targets. These tasks are distinct from other drug discovery datasets as this benchmark is focused directly on surrogate model performance over the baseline computational drug discovery method of docking. A different approach to applying machine learning to docking is the use ML models a scoring function rather than the result of the optimization of the ligand conformation/position relative to the scoring function [19]. Other benchmarks are available to address to gap between docking and experimental binding free energy calculations such as DUD-E [18].

The data has two modes of representation, either 2D or 3D, where the 3D data a ligand conformation in an SDF file. 2D ligand data is available in a CSV file containing the molecule's purchasable name, a SMILES string, and it's associated docking score in a particular complex.

The ligands available for each dataset are sorted into three categories ORD (orderable compounds from Mcule [14]), ORZ (orderable compounds from Zinc [25]), and an aggregate collection which contains all the available compounds plus others (Drug Bank [29], and Enamine Hit Locator Library [9]). Docking failures were treated as omissions in the data, which may be important consideration though typically the number of omissions accounts for 1-2% at most of each sample.

The data is available here, `https://doi.org/10.26311/BFKY-EX6P`, and more information regarding persistence and usage is available on the data website and SI [7].

## 4 Method

At a high level, the use of surrogate models for protein-docking ligand aims to accelerate virtual ligand screening campaigns. A surrogate model aims to replace the CPU-bound docking program with a trained model to filter incoming ligands such that the number of actual docking calculations is minimized and the number of missed compounds due to model accuracy is minimized. In other words, a surrogate model is trained and a cut-off is specific, say 1%. The model is run over the proposed library to screen, and the top 1% of ligands are then docked utilizing the program in order to have the exact scores and pose information as if typically docking. In this way, we do not see current surrogate models as a replacement for docking, but rather as a means of expanding it's use over large virtual libraries. We call this model SPFD, surrogate model prefilter then dock. This model has a single hyperparameter, $\sigma$ which determines after running the surrogate model over the library, what percentage of most promising predicted compounds do we actually dock utilizing traditional docking techniques.

### 4.1 Data frame construction

We used the protein-ligand docking results between the prepared receptors and compounds in the OZD library to build machine learning (ML) data-frames for each binding site. The raw docking scores (the minimum Chemgauss4 score over the ensemble of conformers in a ligand-receptor docking simulation) were processed. Because we were interested in determining strong binding molecules (low scores), we clipped all positive values to zero. Then, since we used the ReLu activation function at the output layer of the deep neural network, we transformed the values to positive by taking the absolute value of the scores. The processed docking scores for each compound to each binding site then served as the prediction target. The code for model training can be found here: `https://github.com/2019-ncovgroup/ML-Code`.

The features used to train the models were computed molecular descriptors. The molecular descriptors were computed as described by Babuji et al [2020]. The full set of molecular features contains 2-D and 3-D descriptors, resulting in a total of 1,826 descriptors. The approximately 6 million docking scores per receptor and 1,826 descriptors were then joined into a data frame for each receptor.

### 4.2 Learning curves

We performed learning curve analysis with the 3CLPro receptor to determine the training behavior of the model. A subset of 2M samples were obtained from the full set of 6M samples. The 2M sample dataset was split into train (0.8), validation (0.1), and test (0.1) sets. We trained the deep neural network on subsets of training samples, starting from 100K and linearly increasing to 1.6M samples (i.e., 80% of the full 2M set). Each model was trained from scratch and we used the validation set to trigger early stopping and the test to calculate measures of generalization performance such as the mean absolute error of predictions. See SI

### 4.3 Model Details

The model was a fully connected deep neural network with four hidden layers each followed by a dropout layer. The dropout rate was set to 0.1. Layer activation was done using the rectified linear unit activation function. The number of samples per gradient update (batch size) was set to 32. The

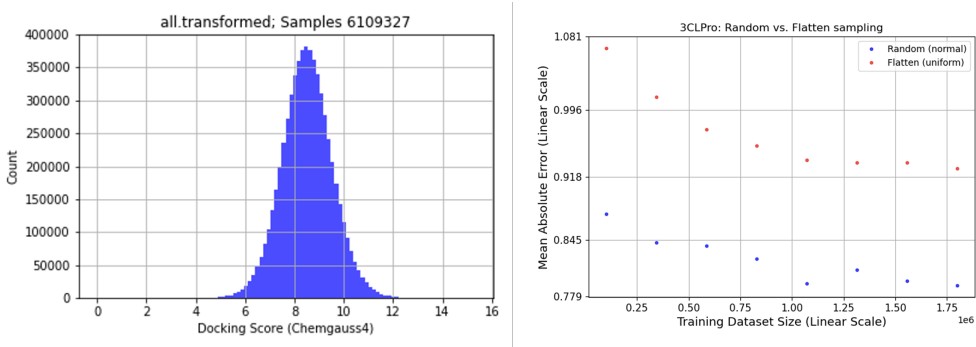

Figure 2: (left) **Histogram of protein-ligand docking of transformed docking scores for 3CL-M$_{pro}$**. The distribution is from the ORZ dataset based on the transformed 2D scores. (right) **Learning curve between dataset size and MAE between random and flattened datasets.**

model was compiled using mean squared error as the loss function and stochastic gradient descent with an initial learning rate of 0.0001 and momentum set to 0.9 as the optimizer. The implementation was python using Keras.

The model was trained by setting the initial number of epochs to 400. A learning rate scheduler was used that monitored the validation loss and reduced the learning rate when learning stagnated. The number of epochs with no improvement after which the learning rate was reduced (patience) was set to 20. The factor by which the learning rate will be reduced was set to 0.75 and the minimum allowable learning rate was set to 0.000000001. Early stopping was used to terminate training If after 100 epochs the validation loss did not improve.

Features were standardized by removing the mean and scaling to unit variance before the onset of training using the entire data frame (before the data frame was split into train and test partitions). The train and test partitions were based on a random 80:20 split of the input data frame. Hyperparameter optimization was performed (see SI.7.7).

Inferencing was performed on Summit. The input was converted to Feather files using the python package feather, which is a wrapper around pyarrow.feather (see the Apache Arrow project at apache.org). Feather formatted files as input in our experience are read faster from disk than parquet, pickle, and comma separated value formats.

## 5    Results

### 5.1    Identification of protein targets and binding receptors

A total of thirty one receptors representing 9 SARS-CoV-2 protein conformations were prepared for docking. These are illustrated in Figure 2 and listed in Table 1. The quality of the receptors reflect what was available at the time the receptor was prepared. For example, whereas the NSP13 (helicase) structure in Table 1was based on homology modeling, today there exists x-ray diffraction models.

### 5.2    Generation of training data

Compounds in the OZD library were docked as described in SI 7.4. The results for the 3CLPro receptor demonstrate a normal distribution (Fig. 2). The best docking scores would be in the range of 12 to 18. The distribution of docking scores for the 3CLPro receptor is illustrative of the distributions for all the other receptors. As shown in the figure, there are very few samples with good docking scores relative to the entire set of samples.

### 5.3    Sampling comparisons

We constructed a set of data frames to investigate the impact of the number of samples, sampling approach, and the choice of drug descriptors as features. The number of samples was further investigated using learning curves. Because we are interested in predicting docking scores in the tail

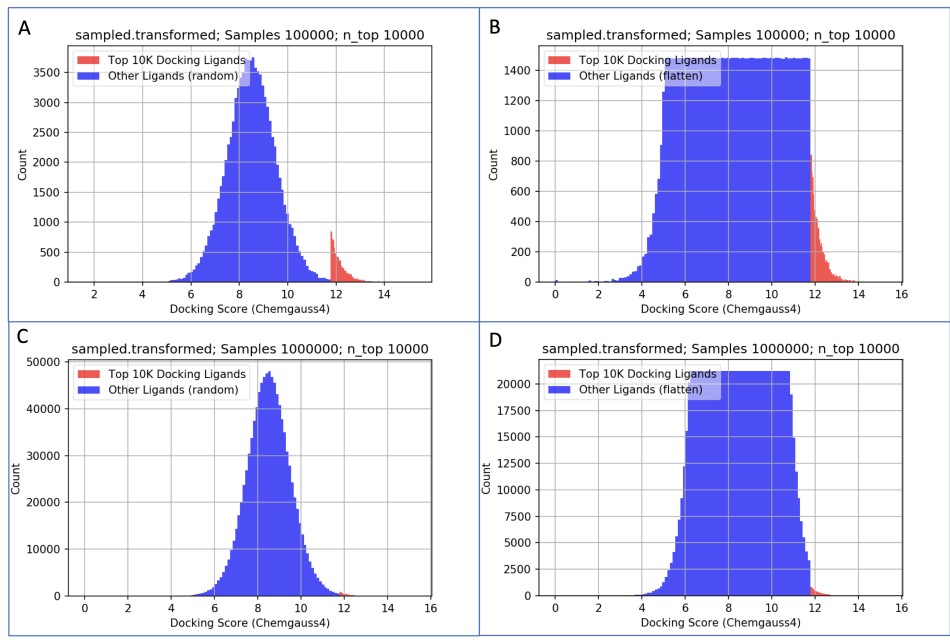

Figure 3: Docking score histograms for each of the four sampling a) 100K-random, b) 100K-flatten, c) 1M-random and d) 1M-flatten approaches used to generate a subset by sampling the full dataset of available scores (approximately six million samples).

of the distribution where the best docking scores exist, we explored two sampling approaches. Lastly, we investigated the impact of using the Mordred 3-D descriptors as features of the compounds.

| Dataset | Count (samples) | Sampling method | Distribution (approximate) |
|---|---|---|---|
| 100K-random | 100,000 | Random | Normal |
| 100K-flatten | 100,000 | Flatten | Uniform |
| 1M-random | 1,000,000 | Random | Normal |
| 1M-flatten | 1,000,000 | Flatten | Uniform |

Table 1: The four sampling approaches used to subset the approx. 6M docking scores for OZD.

We generated a dataset subset by sampling the approximately 6M samples in the OZD data complete data-frames. We examined four sampling approaches, differing by two parameters, as listed in Table 1: (1) the total number of samples drawn from the entire dataset (i.e., the count), and (2) the algorithm used to draw the samples (i.e., the sampling method).

Drawing samples at random preserves the original normal-shaped distribution (thus, the name Random). Alternatively, for a more balanced dataset, we sample scores with an alternative algorithm to create a roughly flattened, uniform-like distribution. To include the highly significant, top score samples, we retain the top ten thousand binding ligands. Figure 3 shows the histograms of the docking scores subset with each of the four sampling scenarios for 3CL-$M_{pro}$. The top ten thousand binding ligands are indicated in red. Note that the distribution of the full dataset can be roughly modeled as a normal distribution, as shown in Figure 2.

When examining the impact of including the Mordred 3-D descriptors in the feature set, we average the validation loss, validation MAE, and validation $r^2$ across the 31 models as we are interested in the aggregate performance of the models across the 31 receptors. Our analysis of the inclusion of the Mordred 3D descriptors is presented in SI Table 3. Our results show no significant advantage to including the 3D descriptors. The results show small improvements in the validation loss across all training data frames when using only the 2D descriptors, and the results are mixed when considering validation $r^2$, with two smaller data frames performing slightly better, and the two larger data frames performing slightly worse. While we do not consider the differences in most cases to be significant,

we do demonstrate that adding the extra training parameters in the form of 3D descriptors does not improve the training performance of the model.

When examining the impact of both the training set size (1M or 100K) and sample selection from either a random distribution or flattened distribution, we average the validation loss, validation MAE, and validation $r^2$ for each trained receptor model that represents one of the thirty one different protein pockets. SI Table 3 shows the differences between the means. A negative value for the validation loss and validation MAE differences would indicate 1M samples achieved a higher quality model, and a positive value for the validation $r^2$ difference would indicate 1M samples achieved a higher quality model. The results indicate that 1M samples from a flattened distribution perform better than 100K samples for all three metrics, whereas 1M samples from a random distribution achieved better metrics for the validation loss and MAE, however the 1M samples from a random distribution had a lower validation $r^2$.

To better understand the differences between the 1M data sets, the Pearson correlation coefficient was calculated between predicted and the observed values from the validation set for each pocket model. In the case of the v5.1-1M samples, the validation set had 200,000 samples. The mean of the PCC across the set of pocket models was calculated for each 1M data set and the V5.1-1M-random is 0.853 and the V5.1-1M-flatten is 0.914.

## 5.4 Learning curve analysis

To further explore the optimal sample size, we generated learning curves for the 3CLPro receptor model and assume 3CLPro will be indicative of other receptors. Using the entire dataset, which contains approximately 6M samples, imposes a significant computational burden for training a deep neural network model for each receptor and performing HP tuning. Regardless of the learning algorithm, supervised learning models are expected to improve generalization performance with increasing amounts of high-quality labeled data. At a certain sample size, however, the rate of model improvement achieved by adding more samples significantly diminishes. The trajectory of generalization performance as a function training set size can be estimated using empirical learning curves.

The range at which the learning curve starts to converge indicates the sample size at which the model starts to exhaust its learning capacity. Figure 2 shows the learning curve where the mean absolute error of predictions is plotted versus the training set size. The curve starts to converge at approximately 1M samples, implying that increasing sample size beyond this range is not expected to improve predictions.

## 5.5 Model Accuracy

The observations of the data frame comparisons (sec. 4.2.3) and learning curves (sec. 4.2.4), specifically:

- The 1613 MOrdred descriptors performed better than the 1826 features in most cases

- The 1M data frames performed better than the 100K data frames in most cases

- The mean Pearson correlation coefficient of the 1M-flatten was higher than that of the 1M-random data frame

led to the selection of training data that was used to create the models listed in SI table 4. These models were then used in inferencing to help guide the selection of compounds for use in whole cell viral inhibition assays.

## 5.6 Inference across 3.8 billion compounds

We divided the 4 billion compounds into 4 input data sets to enable better utilization of resources. ENA, G13, ZIN, OTH. We also constructed a set of compounds from the MCULE data set that could be easily purchased (organic synthesis already done). The MCULE subset was named ORD. The inferencing rate was approximately 50,000 samples per second per GPU, and all 6 GPUs per summit node were used.

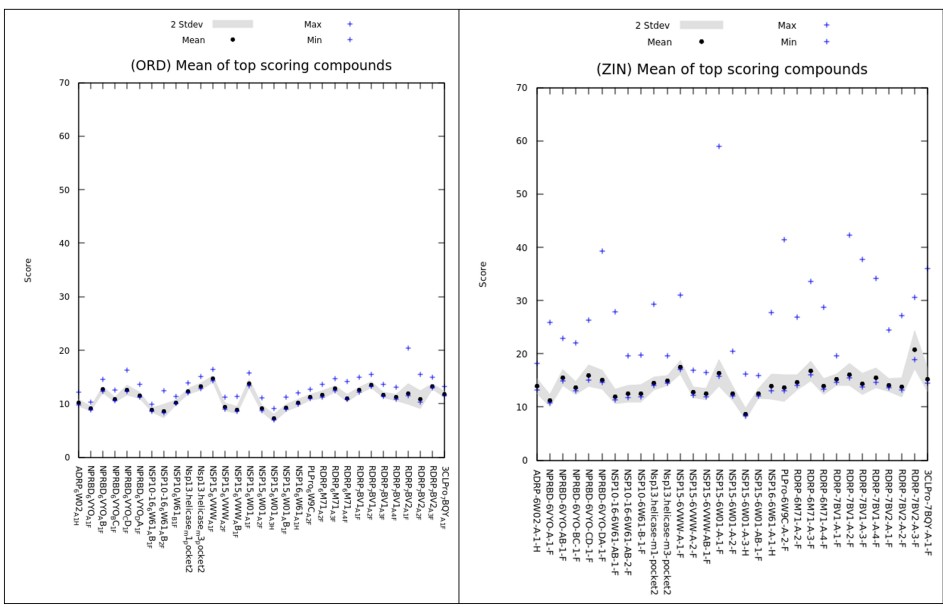

Figure 4: Comparison of the 31 receptor models with the 2000 best scoring compounds from ORD and ORZ.

We analyzed the results for each receptor by selecting the top 2000 scoring compounds and computing mean, standard deviation, maximum, and minimum values. We present two examples of these results in Figure 4. It is interesting to note that the range represented by the maximum and minimum predicted scores for the best 2000 scoring compounds is remarkably different between these two, and in fact ZIN was representative of the others (G13, ENA, and OTH). One working hypothesis is that the compounds in ORD have been shown to be synthesizable, whereas compounds in the other sets are not necessarily synthesizable as these are virtual libraries.

## 5.7 Deep Learning Inference Speed

We discuss the relative speedup of utilizing a pre-filter surrogate model for docking campaigns against a traditional docking campaign. We define two workflows for performing protein-ligand docking over a library of compounds: D (traditional docking, no surrogate-prefilter) and SPFD (surrogate-prefilter then dock). We construct timing models of both of these workflows to understand the relationship between computational accuracy, computational performance (time and throughput), and pre-filter hyperparameter ($\sigma$). We distinguish between the surrogate model's accuracy, which pertains to how well the surrogate model fits the data, and workflow accuracy, which pertains to how well the results of the whole SPFD workflow compares to the results of just traditional docking workflow.

As discussed in Section 4, the choice of pre-filter hyperparameter is a decision about workflow accuracy for detecting top leads. Model accuracy influences the workflow accuracy, but the workflow accuracy can be adjusted with respect to a fixed model accuracy (see figure 5 where the vertical lines each correspond to the same underlying model with fixed accuracy but differentiate the overall workflow accuracy with respect to traditional docking). Therefore we can interpret $\sigma$ as a trade-off between the workflow throughput and workflow accuracy. For example, $\sigma = 1$ is always 100% workflow accurate since traditional docking is run on the whole library when $\sigma = 1$, but $\sigma = 1$ is even slower than traditional docking as it implies docking the whole library as well as utilizing the surrogate model. We can determine the overall workflow accuracy with the model by looking at the RES plot, which is of course has as a factor the performance characteristics of the surrogate model. Given a particular model's accuracy versus performance characteristics, different levels of pre-filtering ($\sigma$), correlate to different tolerances to detecting top-scoring compounds.

For the following analysis, we fix node types for simplicity. Assume the traditional protein-ligand docking software has a throughput $T_D$ in units $\frac{\text{samples}}{(\text{seconds})(\text{node})}$, and the surrogate models have a throughput $T_{SPF}$. Let $t_D$ and $t_{SPFD}$ be the wall-clock time of the two workflows. The time of the traditional

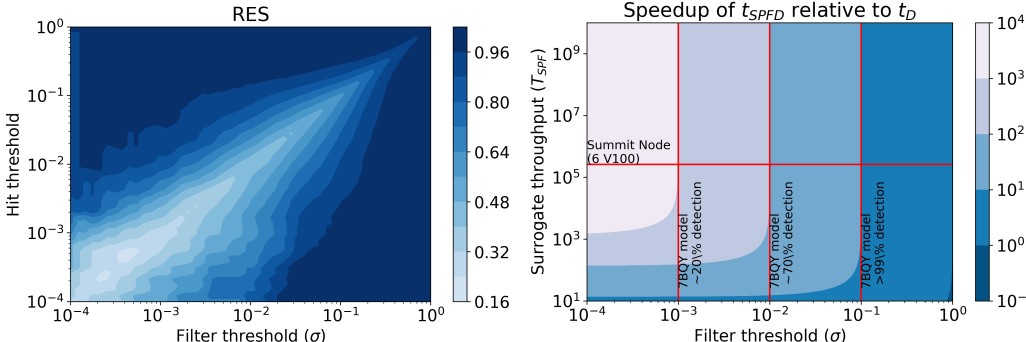

Figure 5: (left) Regression enrichment surface ($n = 200,000$) based on the surrogate model for 7BQY [5]. The $x$-axis represents $\sigma$ which determines the level of filtering the model is used for (i.e., after predicting over the whole library, what percentage of compounds then used in the next stage docking). The $y$-axis is the threshold for determining if a compound is a hit or not. The point $(10^{-1}, 10^{-3})$ is shaded with 100% detection. This implies the model over a test set can filter out 90% of compounds without ever missing a compound with a score in the $10^{-3}$ percentile. In concrete numbers, we can screen 200,000 compounds with the model, take the top 20,000 based on those inference scores, and dock them. The result is running only 20,000 docking calculations, but those would contain near 100% of the top 200 compounds (as if one docked the entire dataset). (right) Based on equation (1) we compute the relative speedup using surrogate models over traditional workflows with fixed parameters library size (1 billion compounds) and $T_D = 1.37 \frac{\text{samples}}{\text{seconds node}}$. The horizontal line indicates where current GPU, surrogate model, throughput is, $T_{\text{SPF}}$, and the vertical lines correspond to the RES plot values for hit threshold equal to $10^{-3}$. The right-most vertical line implies a VLS campaign with surrogate models where the surrogate GPU-based model can with accuracy $> 99\%$ detect the top 10% from the bottom 90% implying a 10x speedup over traditional methods. By adding surrogate models as a pre-filter to docking, scientists can dock 10x more in the same amount of time with little detectable loss.

workflow, $t_D$, and the time of the surrogate prefilter then dock workflow, $t_{\text{SPFD}}$, are

$$t_D = \frac{L}{T_D} \quad \text{and} \quad t_{\text{SPFD}} = \frac{\sigma L}{T_D} + \frac{L}{T_{\text{SPF}}}. \tag{1}$$

Notice, that $t_{\text{SPFD}}$ is simply the sum of the time of running the surrogate model over the library, $L/T_{\text{SPF}}$, and the time of traditional docking the highest scoring $\sigma L$ compounds.

$$\text{Speedup} = \frac{T_{\text{SPF}}}{T_D + \sigma T_{\text{SPF}}} \tag{2}$$

This implies that the ideal speedup of our workflow is directly dependent on the throughput of both the docking calculation, surrogate model, and the parameter $\sigma$. $\sigma$ is indirectly dependent on the model accuracy. If the surrogate model was completely inaccurate, even though $\sigma = 10^{-3}$ implies a 1000x speed up, no hits would be detected. If one wants to maximize workflow accuracy, that is not miss any high scoring compounds compared to traditional docking, then they must supply a threshold for hits (corresponding to the $y$-axis of RES). Suppose this threshold is $y_{\text{thres}} = 10^{-3}$. If they wanted to maximize not missing any compounds, they should set $\sigma$ to $10^{-1}$ based on this model's RES plot since that is the smallest value of $\sigma$ such that the detection accuracy of the surrogate model is near 100%. But, it is not always the case downstream tasks require 100% detection—hence $\sigma$ is a true hyperparameter.

We infer $T_{\text{SPF}}$ on a Summit (Oak Ridge Leadership Computing Facility) and on an A100 ThetaGPU node (Argonne Leader Computing Facility). Both tests were using 64 nodes, 6 GPUs per node, but the throughput was computed per GPU. We found the V100 summit node was capable of 258.0K $\frac{\text{samples}}{\text{(s)(node)}}$ while the A100 nodes were 713.4K $\frac{\text{samples}}{\text{(s)(node)}}$. We infer $T_D$ as 1.37 $\frac{\text{samples}}{\text{(s)(node)}}$ based on a CPU docking run over 4,000 summit nodes with 90% CPU utilization from [6]. Thus, we can compute the speedup based on a Summit node head-to-head comparing setting $T_{\text{SPF}}$ to 258.0K and $T_D$ to 1.37 in Equation

1, resulting in a speedup of 10X for $\sigma = 0.1$. Based on the RES analysis in Figure 5, $\sigma$ of 0.1 corresponds to a model accuracy of near 100% for filtering high scoring leads ($> 1\%$ of library). If one is willing to trade-off some loss of detection, say 70% detection of high scoring leads, then the speedup is 100X. The extreme case, a choice of $\sigma = 10^{-3}$, implies a speedup 1000X but means only roughly 20% of the top scoring leads may appear at the end.

The analysis of SPFD implies that speedups are essentially determined by $\sigma$ while $T_{\mathrm{SPF}}$ does not have as large of an effect (this is based on how fast ML inference currently is). As a hyperparameter, $\sigma$ is dependent on the workflow's context and, in particular, what the researchers are after for that virtual ligand screening campaign. We can say, though, at least informally, model accuracy and $\sigma$ are highly related. The more accurate the models are, the better the RES plot gets as one is willing to trust the ML model for filtering the best compounds from the rest. In figure 5, the $x$-axis of both plots are similar. The accuracy of a particular $\sigma$ is found by setting one's level of desired detection, the $y$-axis of the RES plot, and then checking the $(\sigma, y)$ point to see how accurate the model is there. The choice of $\sigma$ is subjective based on how accurate one needs the model for their $y$-axis threshold for hits. We focus mainly on the case of no loss of detection, which means $\sigma = 0.1$ for our particular trained models. In order to focus on the theoretical model of relating computational accuracy, confidence (again, in a colloquial sense), and computational performance, we simplify over a richer model of performance analysis assuming uniformity of task timing and perfect scaling. Furthermore, we chose a head-to-head comparison of a particular node type's CPU performance to GPU performance. At the same time, we could have compared the best non-surrogate model workflow times to the best surrogate model workflow times.

# 6   Discussion

We demonstrate an accelerated protein-ligand docking workflow called surrogate model prefiltering then dock (SPFD) which is at least 10x faster than traditional docking with nearly zero loss of detection power. We utilize neural network models to learn a surrogate to the CPU-bound protein-ligand docking code. The surrogate model has a throughput over six orders of magnitude than the standard docking code. By combining these workflows, utilizing the surrogate model as a prefilter, we can gain a 10x speedup in the traditional docking software without losing any detection ability (for hits defined as the best scoring 0.1% of a compound library). We utilize regression enrichment surfaces to perform this analysis. The regression enrichment surface plot is more illustrative than the typical accuracy metrics reported from deep learning practices. Figure 5 showcases our initial models at this benchmark show a 10x speedup without loss of detection (or 100x speedup with 70% detection). This 10x speedup means if a current campaign takes one day to run on library size $L$, in the same amount of time without missing leads, one can screen ten times as many compounds. Given the potential for 100x or even 1000x speedup for docking campaigns, we hope to advance the ability of surrogate models to filter at finer levels of discrimination accurately.

Utilizing SPFD we report a 10x speedup to traditional docking with little to no loss of accuracy or methodical changes needed besides just utilizing ML-models as a prefilter. On the other hand, we see these results as conceptually tying together the model application (hit threshold and adversity to detection loss, choice of $\sigma$) with more traditional analysis such as performance characteristics and model performance evaluation. Our results of the timing analysis also imply a need for more accurate models that are more sensitive to screening large libraries. Our analysis, outlined in Figure 5, highlight the choice of prefilter threshold as the limiting factor for seeing orders of magnitude speedup. In particular, focusing on speedups which show *no loss* of detection, model accuracy must be pressed forward as there is no path to accelerating traditional docking workflows without more accurate surrogate models. Given out of box modeling technique can speed up virtual screening 10x with no loss of detection power for a reasonable hit labeling strategy (top 0.1%), we believe the community is not far from 100x or even 1000x. The way to get there is to boost our model accuracies or develop techniques to recover hits in lossy SPFD regimes (such as not improving model performance but decreasing $\sigma$ to $10^{-2}$ and applying another technique to recover the 30% loss of detection power). We believe this benchmark is important, as successful early drug discovery efforts are essential to rapidly finding drugs to emerging novel targets. Improvements in this benchmark will lead to orders of magnitude improvement in drug discovery throughput.

## Acknowledgements

This research was supported by the Exascale Computing Project (17-SC-20-SC), a collaborative effort of the U.S. Department of Energy Office of Science and the National Nuclear Security Administration.

We acknowledge other members of the National Virtual Biotechnology Laboratory (NVBL) Medical Therapeutics group. We acknowledge computing time via the COVID19 HPC Consortium.

Research was supported by the DOE Office of Science through the National Virtual Biotechnology Laboratory, a consortium of DOE national laboratories focused on response to COVID-19, with funding provided by the Coronavirus CARES Act. SJ also acknowledges support from ASCR DE-SC0021352.

*Scientific and Technical Information Only For All Information.*
Unless otherwise indicated, this information has been authored by an employee or employees of UChicago Argonne, LLC., operator of Argonne National Laboratory, with the U.S. Department of Energy. The U.S. Government has rights to use, reproduce, and distribute this information. The public may copy and use this information without charge, provided that this Notice and any statement of authorship are reproduced on all copies. While every effort has been made to produce valid data, by using this data, User acknowledges that neither the Government nor operating contractors of the above national laboratories makes any warranty, express or implied, of either the accuracy or completeness of this information or assumes any liability or responsibility for the use of this information. Additionally, this information is provided solely for research purposes and is not provided for purposes of offering medical advice. Accordingly, the U.S. Government and operating contractors of the above national laboratories are not to be liable to any user for any loss or damage, whether in contract, tort (including negligence), breach of statutory duty, or otherwise, even if foreseeable, arising under or in connection with use of or reliance on the content displayed in this report.

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

# 7 Dataset Details and Use

## 7.1 Persistence, Ethics, and License

The 2D version of the dataset is hosted publicly by a third party provider, FigShare for redundancy (for which they determine persistence), https://doi.org/10.6084/m9.figshare.14745234. The rest of the data, including all the 3D structures, is hosted by Argonne's Leadership Computing Center and accessible via a Globus endpoint with documentation hosted by GitHub. The authors are confident the data will be persistent across FigShare, GitHub, ALCF, and Globus.

The authors believe this data presents a minimal ethical risk to the community at large. The proposed dataset contains computer-generated protein-ligand structures and computed scores. Information of this sort, albeit at a smaller scale, is widely available on the web currently, and releasing this particular dataset would not set any new standards (in terms of a qualitative assessment of data type). The authors believe the biggest risk of releasing this dataset would be localized to one's interpretation of resulting models, theories, or endeavours based on inductive reasoning from the data alone—but, this risk is typical of any scientific dataset.

## 7.2 Data Generation Methods

The original data was generated by Open Eye Scientific's FRED docking protocols, and was aggregated, cleaned, and standardized with naming convetions. The code was generated with this software: github.com/inspiremd/Model-generation.

### 7.3 Docking Protocol

The training and testing datasets for these experiments were generated using 31 protein receptors, covering 9 diverse SARS-CoV-2 viral target protein conformations, that target (1) 3CLPro (main protease, part of the non-structural protein/ NSP-3), (2) papain like protease (PLPro), (3) SARS macrodomain (also referred to as ADP-ribosyltransferase, ADRP), (4) helicase (NSP13), (5) NSP15 (endoribonuclease), (6) RNA dependent RNA polymerase (RDRP, NSP7-8-12 complex), and (7) methyltransferase (NSP10-16 complex). For each of these protein targets, we identified a diverse set of binding sites along the protein interfaces using two strategies: for proteins that had already available structures with bound ligands, we utilized the X-ray crystallography data to identify where ligand densities are found and defined a pocket bound by a rectangular box surrounding that area; and for proteins that did not have ligands bound to them, we used the FPocket toolkit that allowed us to define a variety of potential binding regions (including protein interfaces) around which we could define a rectangular box. This process allowed us to expand the potential binding sites to include over 90 unique regions for these target proteins. We use the term target to refer to one binding site. The protocol code can be found here: `https://github.com/2019-ncovgroup/HTDockingDataInstructions`.

### 7.4 Preparation of Ligand Libraries

Two ligand libraries were prepared. The first was the orderable subset of the Zinc15 database (we refer to this as OZD) and the second was the orderable subset of the MCULE compound database (we refer to this as ORD). The generation of the orderable subsets was primarily a manual activity that involved finding all compounds that are either in stock or available to ship in three weeks across a range of suppliers. Consistent SMILE strings and drug descriptors for the orderable subsets of the Zinc15 and MCULE compound databases were generated as described by Babuji et al [2020]. Drug descriptors for the Zinc15 and MCULE compound databases can be downloaded from the nCOV Group Data Repository at `https://2019-ncovgroup.github.io`.

### 7.5 Docking Protocols

We used OpenEye Toolkits for docking six million (6M) small-molecules from the OZD database. For each ligand from the database, we calculate a single Chemgauss4 score as a surrogate for binding affinity. For each ligand in the database (provided as a SMILES string), we create an ensemble of structures, sampled from various proteinization states (tautomers) and 3-D conformers. Typically, this results in approximately one thousand 3-D structures for each ligand. If the ligand's stereochemistry is ambiguous from the provided SMILES, enantiomers are enumerated prior to conformer generation. Each conformer is then docked to the protein target using FRED or HYBRID depending on the availability of a bound ligand in the crystal structure of the specific target. Scores are calculated over the best pose over the ligand ensemble using the Chemgauss4 scoring function. The scoring function is unitless and aims to measure the fitness of a ligand pose in an active site via a numerical score. Poses with more negative scores are more likely to be correctly docked. In order to produce a single score for each database SMILES entry, the minimum over the ensemble is returned. The typical range of the scoring function is between -20 and 0, though the scoring function range is unbounded.

### 7.6 Data Cleanliness

Failure analysis of docking runs is available in the SI of [6].

### 7.7 Model Hyperparameter Optimization

The CANDLE framework was used subsequently used to tune the deep neural network for future training and screening activities. The CANDLE compliant deep neural network was tuned in two phases. The first involved using two CANDLE hyperparameter optimization workflows - mrlMBO and GA. Each differs in the underlying ML techniques used to optimize the hyperparameters. The second phase involved implementing and testing new sample weighting strategies in an attempt to weight the samples at the good end of the distribution more heavily during training. Results of the GA and mlrMBO workflows produced a model architecture that had a 6.6% decrease in the validation mean absolute error and a 2.8% increase in the validation R-squared metrics.

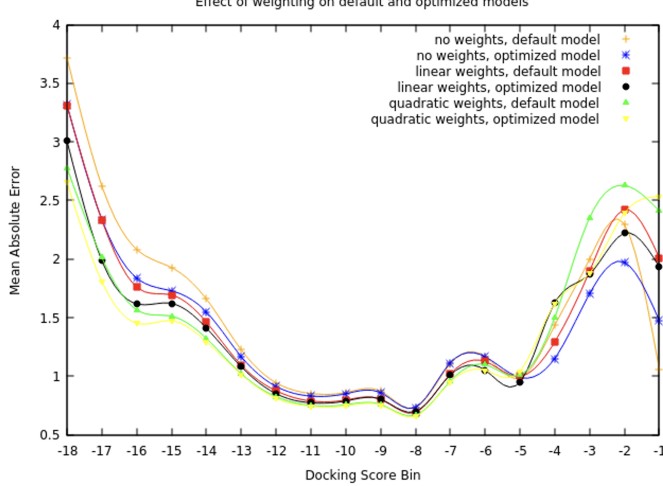

Figure 6: (left) Effects of sample weighting strategies on the default and optimized model.

Efforts to decrease the error in the good tail of the distribution (where the docking scores are best) focused on adding sample weights to the model while training. We investigated linear and quadratic weighting strategies. We applied the weighting strategies to both the default model as well as the hyperparameter optimized model. The linear strategy weights the sample proportionally with the docking score, while the quadratic scales with the square of the docking score. These strategies generic in that they can be applied to basically any training target value. To analyze the impact of the weighting strategies, we computed the mean absolute error on bins of predicted scores with a bin interval of one. These results are presented in Figure 6.

## 7.8 Model Scores

See table 2.

## 7.9 Modeling Feature Details

| pocket (model) | epochs | loss | mae | r2 | val_loss | val_mae | val_r2 |
|---|---|---|---|---|---|---|---|
| 3CLPro_7BQY_A_1_F | 513 | 0.338 | 0.454 | 0.870 | 0.426 | 0.505 | 0.838 |
| ADRP_6W02_A_1_H | 599 | 1.020 | 0.782 | 0.786 | 1.326 | 0.876 | 0.724 |
| NPRBD_6VYO_A_1_F | 453 | 0.302 | 0.427 | 0.848 | 0.356 | 0.466 | 0.822 |
| NPRBD_6VYO_AB_1_F | 599 | 0.482 | 0.540 | 0.800 | 0.601 | 0.597 | 0.752 |
| NPRBD_6VYO_BC_1_F | 427 | 0.566 | 0.586 | 0.899 | 0.702 | 0.653 | 0.876 |
| NPRBD_6VYO_CD_1_F | 523 | 0.474 | 0.534 | 0.816 | 0.602 | 0.597 | 0.769 |
| NPRBD_6VYO_DA_1_F | 587 | 0.485 | 0.541 | 0.854 | 0.591 | 0.595 | 0.824 |
| NSP10-16_6W61_AB_1_F | 283 | 0.490 | 0.546 | 0.902 | 0.615 | 0.613 | 0.878 |
| NSP10-16_6W61_AB_2_F | 387 | 0.523 | 0.565 | 0.901 | 0.655 | 0.628 | 0.877 |
| NSP10_6W61_B_1_F | 433 | 0.576 | 0.590 | 0.908 | 0.677 | 0.631 | 0.893 |
| Nsp13.helicase_m1_pocket2 | 338 | 0.553 | 0.577 | 0.867 | 0.663 | 0.633 | 0.843 |
| Nsp13.helicase_m3_pocket2 | 434 | 0.485 | 0.538 | 0.878 | 0.582 | 0.585 | 0.855 |
| NSP15_6VWW_A_1_F | 406 | 0.526 | 0.563 | 0.837 | 0.640 | 0.621 | 0.804 |
| NSP15_6VWW_A_2_F | 441 | 0.336 | 0.451 | 0.876 | 0.417 | 0.506 | 0.849 |
| NSP15_6VWW_AB_1_F | 599 | 0.234 | 0.376 | 0.829 | 0.298 | 0.423 | 0.784 |
| NSP15_6W01_A_1_F | 596 | 0.473 | 0.533 | 0.835 | 0.595 | 0.597 | 0.795 |
| NSP15_6W01_A_2_F | 451 | 0.313 | 0.434 | 0.888 | 0.378 | 0.475 | 0.865 |
| NSP15_6W01_A_3_H | 530 | 0.759 | 0.679 | 0.784 | 0.967 | 0.754 | 0.727 |
| NSP15_6W01_AB_1_F | 470 | 0.261 | 0.396 | 0.829 | 0.316 | 0.434 | 0.796 |
| NSP16_6W61_A_1_H | 583 | 1.044 | 0.795 | 0.787 | 1.339 | 0.888 | 0.728 |
| PLPro_6W9C_A_2_F | 512 | 0.343 | 0.458 | 0.858 | 0.427 | 0.508 | 0.825 |
| RDRP_6M71_A_2_F | 461 | 0.311 | 0.430 | 0.855 | 0.384 | 0.479 | 0.823 |
| RDRP_6M71_A_3_F | 498 | 0.495 | 0.548 | 0.859 | 0.599 | 0.602 | 0.830 |
| RDRP_6M71_A_4_F | 463 | 0.382 | 0.481 | 0.837 | 0.465 | 0.528 | 0.803 |
| RDRP_7BV1_A_1_F | 394 | 0.312 | 0.433 | 0.853 | 0.378 | 0.477 | 0.823 |
| RDRP_7BV1_A_2_F | 531 | 0.497 | 0.546 | 0.848 | 0.603 | 0.597 | 0.817 |
| RDRP_7BV1_A_3_F | 451 | 0.453 | 0.524 | 0.849 | 0.550 | 0.583 | 0.818 |
| RDRP_7BV1_A_4_F | 420 | 0.385 | 0.481 | 0.873 | 0.476 | 0.536 | 0.844 |
| RDRP_7BV2_A_1_F | 589 | 0.304 | 0.428 | 0.821 | 0.369 | 0.469 | 0.784 |
| RDRP_7BV2_A_2_F | 422 | 0.441 | 0.516 | 0.839 | 0.562 | 0.581 | 0.798 |
| RDRP_7BV2_A_3_F | 510 | 0.466 | 0.531 | 0.830 | 0.579 | 0.590 | 0.791 |

Table 2: Table of released model's training details and validation scores. Released model files and corresponding code is avilable from the project GitHub.

1613 Features

| Model | epoch | val loss | val MAE | val $r^2$ |
|---|---|---|---|---|
| V5.1-100K-flatten-2 | 337 | 0.80 | 0.66 | 0.71 |
| V5.1-100K-random-2 | 336 | 0.80 | 0.66 | 0.71 |
| V5.1-1M-flatten-2 | 484 | 0.60 | 0.59 | 0.81 |
| V5.1-1M-random-2 | 455 | 0.49 | 0.52 | 0.68 |

1826 Features

| Model | epoch | val loss | val MAE | val $r^2$ |
|---|---|---|---|---|
| V5.1-100K-flatten-2 | 313 | 0.97 | 0.74 | 0.85 |
| V5.1-100K-random-2 | 330 | 0.81 | 0.67 | 0.71 |
| V5.1-1M-flatten-2 | 462 | 0.60 | 0.59 | 0.81 |
| V5.1-1M-random-2 | 456 | 0.52 | 0.54 | 0.67 |

Table 3: Impact of including Mordred 3-D descriptors in the training data for the different sampling strategies.

| Sample Selection Strategy | epoch | val loss | val mae | val r2 |
|---|---|---|---|---|
| V5.1-100K-flatten | 337 | 0.80 | 0.66 | 0.71 |
| V5.1-100K-random | 336 | 0.80 | 0.66 | 0.71 |
| V5.1-1M-flatten | 484 | 0.60 | 0.59 | 0.81 |
| V5.1-1M-random | 455 | 0.49 | 0.52 | 0.68 |

Difference of 1M Samples - 100K Samples

| Sample Selection Strategy | $\Delta$ epoch | $\Delta$ val loss | $\Delta$ val mae | $\Delta$ val r2 |
|---|---|---|---|---|
| flatten | 147 | -0.20 | -0.07 | 0.11 |
| random | 119 | -0.31 | -0.14 | -0.03 |

Table 4: Comparison of 1M samples to 100K samples.

