# OpenReview forum: "Protein-Ligand Docking Surrogate Models: A SARS-CoV-2 Benchmark for Deep Learning Accelerated Virtual Screening"
_NeurIPS.cc/2021/Track/Datasets_and_Benchmarks/Round1 — Submitted to NeurIPS 2021 Datasets and Benchmarks Track (Round 1)_

### Official Review · Reviewer_CAot · 2021-07-02
**deep learning surrogate prefiltering**

**Rating:** 5
**Confidence:** 3

**Strengths:**

Interesting benchmark and dataset on a very relevant current topic.

**Weaknesses:**

I struggled understanding a lot of details of this work. I feel many researchers that are not actively in this field will have difficulty figuring out how to actually train a model on this data, the evaluation metrics and the ways to do analysis on the results. The work may benefit from having a co-author that is a Machine learning researcher but not someone working on virtual screening, to ensure that the benchmark and its details are accessible (from an understanding point of view) to everyone.

**Additional Feedback:**

Nothing beyond what's written above.

**Clarity:**

For someone that is not a researcher in this field, the paper is quite inaccessible and uses a lot of jargon that is likely unfamiliar to most people at NeurIPS.

**Correctness:**

I am not an expert enough in this field to assess this. The model training part seems correct.

**Documentation:**

Yes, the documentation appears fine.

**Ethics:**

No concerns.

**Relation To Prior Work:**

I am not an expert enough in this field to assess this.

**Summary And Contributions:**

This is a dataset of 200M 3D and 2D structures. The main idea is to build so-called "surrogate" models (deep nets) that can act as pre-filters: the top-ranked candidates identified by the deep nets are then actually analyzed by the docking software. The work claims that the error rate is very small this way (in terms of detection) and that one can get big speed-ups.

A few comments:

* It's not immediately obvious what Section 3.2 is describing. I assume it's talking about the training/validation/test curve when training the model. If so, those curves may be useful to show in the actual paper.
* The model in Section 3.3 seems very simple: have the authors actually tried different model sizes and structures?
* The sections on whether more training data helps are interesting but perhaps not overall surprising. The conclusion that after 1M the returns are diminishing makes sense, but this fact is only useful for someone that would try to collect such data again, no?
*  I would have expected another kind of analysis in Section 4.7: as I understand it, \sigma acts as a threshold of sorts and it would make sense to me to have either a precision-recall or ROC curve in this case, wouldn't you agree? Perhaps Figure 4 has this, but I am unfamiliar with such curves so it was hard to understand.

---

### Official Review · Reviewer_tLVB · 2021-07-04
**Very interesting benchmark and utility for ML methods. This will encourage interesting research for drug discovery applications.**

**Rating:** 8
**Confidence:** 3
**Clarity:** Yes the paper is very well written an…

**Strengths:**

1. The paper is well written and the utility of ML is well motivated
2. Authors have clearly and rigorously demonstrated the utility as a screening tool.
3.  I think this benchmark can really open up an interesting challenge for ML researchers for a very important and timely scientific application

Based on the author response and other reviews, I would like to keep the score as is.

**Weaknesses:**

The main weakness I see is a little more detailed breakdown (for non-domain experts like me) of the errors made and contextualizing it for the application at hand. I think this is fixable with some added description of what is acceptable and what is not (what is the risk tolerance of  false negatives and how well ML models are performing in that regard)

**Additional Feedback:**

Please clarify the acceptable error rate behavior a bit more for understanding of (and evaluation) for non-domain experts like ML researchers who might use such a benchmark

**Correctness:**

Yes I believe the evaluation and experiment design is thorough, rigorous and clear.

**Documentation:**

Yes the description in the main paper is detailed, and sufficient.

**Ethics:**

I do not see any ethical concerns with such a benchmark and I believe this could be a fairly high utility benchmark for ML researchers.

**Relation To Prior Work:**

The prior work and existing benchmarks are highlighted to the best of my knowledge. However, I am not a domain expert (since I am an ML researcher) and I am relying on authors related work for this comment.

**Summary And Contributions:**

The authors propose to use ML methods as surrogate models for high throughput drug discovery, particularly to screen compounds to receptors of candidate pathogen proteins. These models are not meant to replace existing screening methods, but used as a pre-filter/screening since the actual screening methods are computationally intensive. Instead with quite reasonable error rates (0.1%), ML methods seem to demonstrate utilty for pre-filtering purposes. The authors have demonstrated extensive experimentation on a variety of tasks and very carefully conducted an empirical analysis of the utility.

---

### Official Review · Reviewer_B2H1 · 2021-07-06

**Rating:** 6
**Confidence:** 3
**Clarity:** Yes.

**Strengths:**

Large dataset and task definition with an aim to become a benchmark.


**Weaknesses:**

It is unclear why approximation of an existing utility should become a major benchmark. I would expect more that physical experiments performed would serve as a better basis of a benchmark dataset.

**Additional Feedback:**

None.

**Correctness:**

The claim of speeding up the slower computation by approximating it appears correct.

**Documentation:**

The data appears clean. Compounds are encoded as SMILES which are a common standard.

**Ethics:**

No.

**Relation To Prior Work:**

There is no explicit related work section. There are about 3 paragraphs positioning this paper in the introduction. I am not sure how novel this work is based on my own knowledge.

**Summary And Contributions:**

The paper proposes a dataset, task, as well as an approach to approximate a typically slow computation to compute binding affinity. By using a neural network to approximate this, they are able to gain an impressive computation speed ("10x speedup") with minimal performance loss.

---

### Decision · Program_Chairs · 2021-07-27

**Decision:**

Reject

**Comment:**

The authors introduce a new dataset for studying protein-ligand docking containing 200M 3D structures across the SARS-CoV-2 proteome. The reviewers commend the timeliness of the problem and the rigor of the work. However, they also raise concerns about the accessibility of the paper to a NeurIPS audience that is not familiar with the field. After the rebuttal phase, they still recommend against accepting the paper.